# Genetic characterization for lesion mimic and other traits in relation to spot blotch resistance in spring wheat

Shweta Singh[1], Vinod Kumar Mishra[2]*, Ravindra Nath Kharwar[1], Neeraj Budhlakoti[3], Ram Narayan Ahirwar[2], Dwijesh Chandra Mishra[3], Sundeep Kumar[4], Ramesh Chand[5], Uttam Kumar[6], Suneel Kumar[4], Arun Kumar Joshi[7,8]

**1** Department of Botany, Institute of Science, Banaras Hindu University, Varanasi, India, **2** Department of Genetics and Plant Breeding, Institute of Agricultural Sciences, Banaras Hindu University, Varanasi, India, **3** ICAR-Indian Agricultural Statistics Research Institute, New Delhi, India, **4** ICAR-National Bureau of Plant Genetic Resources, New Delhi, India, **5** Department of Mycology and Plant Pathology, Institute of Agricultural Sciences, Banaras Hindu University, Varanasi, India, **6** Borlaug Institute for South Asia (BISA), Ladhowal, Ludhiana, Punjab, India, **7** International Maize and Wheat Improvement Center (CIMMYT), New Delhi, India, **8** Borlaug Institute for South Asia (BISA), New Delhi, India

* vkmbhu@gmail.com

**Data Availability Statement:** All relevant data are uploaded to the CIMMYT repository and publicly accessible via the following URL: http://hdl.handle.net/11529/10714 (Hdl/11529/10714).

## Abstract

Lesion mimic (Lm) mutants display hypersensitive responses (HR) without any pathogen attack; their symptoms are similar to those produced by a pathogen and result in cell death. In wheat, such mutants have been reported to be resistant against leaf rust due to their biotrophic nature. However, Lm mutants tend to encourage spot blotch (SB) disease caused by *Bipolarissorokiniana* since dead cells facilitate pathogen multiplication. In this study, 289 diverse wheat germplasm lines were phenotyped in three consecutive growing seasons (2012–2015). Genotype data was generated using the Illumina iSelect beadchip assay platform for wheat germplasm lines. A total of 13,589 single-nucleotide polymorphisms (SNPs) were selected andused for further association mapping. Lm was positively associated with Area Under Disease Progress Curve (AUDPC) for SB but negatively with glaucous index (GI), leaf tip necrosis (*Ltn*) and latent period (LP). *Ltn* had a negative association with AUDPC and Lm but a positive one with LP. In a genome-wide association study (GWAS), 29 markers were significantly associated with these traits and 27 were an notated. Seven SNP markers associated with Lm were on chromosome 6A; another on 1B was found to be linked with *Ltn*. Like wise, seven SNP markers were associated with GI; one on chromosome 6A with the others on 6B. Five SNP markers on chromosomes 3B and 3Dwere significantly correlated with LP, while nine SNP markers on chromosomes 5A and 5B were significantly associated with AUDPC for SB. This study is the first to explore the interaction in wheat between Lm mutants and the hemibiotrophic SB pathogen *B.sorokiniana*.

**Funding:** The Author (Shweta Singh) is grateful to the University Grant Commission (UGC), New Delhi for providing JRF, SRF and financial assistance. The funder had no role in study design, data collection and analysis, decision to publish, or preparation of the manuscript.

**Competing interests:** The authors have declared that no competing interests exist.

## Introduction

Plant lesion mimic(Lm) mutants exhibitnecrotic symptoms on the leaves that appear in the absence of any pathogen. These symptoms mimic the HR displayed during plant-pathogen interactions [1]. First reported in barley [2,3], this type of mimicry has also been witnessed in *Arabidopsis* [4–6], maize [7,8], rice [9–11], and wheat [12,13]. Lesion mimics form patches of dead cells without any natural wound, injury, stress or infection being present in the plants. HR-associated cell death arrests the growth of biotrophic pathogens by restricting the supply of essential nutrients from the host. However, it confers no resistance to hemibiotrophs or necrotrophic pathogens [14] such as *Botrytis cinerea* [15], where cell death follows infection [16]. Another trait, leaf tip necrosis (*Ltn*), provides resistance against various plant pathogens at the adult plant stage and is linked to genes such as *Lr34* [17]. This gene was first reported in wheat PI58548 [18] and was later described in many wheat varieties [19–21]. It slows development of rusts and, under suitable conditions, has the ability in seedlings to provide resistance to certain rust races including leaf rust (*Pucciniatriticina*) and stripe rust (*P.striiformis*) [19]. Wheat genotypes possessing *Lr34* also show resistance against SB caused by hemibiotroph pathogen *B.sorokiniana* [17]. SB progress may also be inhibited by other components of resistance such as increased latent period (LP) [17] and glaucous index (GI) or waxiness. Disease potential of the crop may be reduced by utilizing genotypes with a long LP [17] and a high GI.

Wheat SB causes average yield reductions in South Asia and India of 19.6% and 15.5% respectively [22]. Losses may be20–80% in susceptible genotypes [23] and complete failure can occur with the most severe infections [24]. *B.sorokiniana* infects most Poaceae family crops, but even though it can infect a vast range of cultivars including wild and cultivated varieties, the chances of the migration of an isolate from one crop to another are remote because the causal pathogen is mainly seed born [25,26].

Because Lm checks the growth of stem rust pathogen, research using molecular markers has been undertaken to find robust Quantitative Trait Loci (QTLs) against stem rust [27]. This is in contrast to the case of Lm and SB where the phenotype-genotype association is not well understood. To our knowledge associations between Lm, *Ltn*, GI, LP, and SB have not previously been studied.

Because lesion mimics restrict the growth of biotrophic pathogens such as rusts, Lm genes are being introduced into wheat cultivars to achieve a degree of immunity against these pathogens. However, the cell death of leaves in response to the Lm genes expression is a major drawback, as this provides suitable conditions for the growth of hemi biotroph and necrotrophs pathogens. Therefore it is important for wheat-cultivating areas globally to establish the effects of Lm genes on SB, which is caused by a hemibiotrophic fungus.

For complex traits, association mapping can identify significant correlations between phenotypes and the corresponding sequence variants within an existing diversity panel [28]. The present study was undertaken to characterize Lm, *Ltn*, GI, and LP, and their association with SB resistance in spring wheat.

## Materials and methods

### Plant material

The Wheat Association Mapping Initiative (WAMI) panel of 289 diverse wheat germplasm lines was obtained from the Global Wheat Program (CIMMYT, Mexico). It contains a wide range of genotypic and phenotypic genotypes which are stable for the traits under examination. These lines were evaluated for Lm, *Ltn*, LP, GI and AUDPC for SB. The details of the germplasm lines used are given in S3 Table.

## Sowing and maintenance of crop under experimental field

The research trials took place at the Agricultural Research Farm, Institute of Agricultural Sciences, Banaras Hindu University, Varanasi, India (25°15' N, 25°15', 83°03'E; 70 m above sea level) during three consecutive crop seasons, 2013–14, 2014–15 and 2015–16. Planting was done between 26th November and 5th December in each crop season to ensure that grain filling coincided with local high temperatures and relative high humidity. Wheat genotypes were sown in two replicates each year in an alpha lattice design. Each genotype was sown in two 1-metre rows, with a row-to-row distance of 25 cm and a plant-to-plant distance of 5 cm. Agronomic practices recommended for normal fertility conditions for irrigated wheat were followed for all three crops; 120 kg N, 60 kg $P_2O_5$ and 40 kg $K_2O$ ha$^{-1}$.

## Inoculation of the pathogen

A pure culture of *B.sorokiniana* (HD 3069/MCC 1572) for artificial inoculation was obtained from the Department of Mycology and Plant Pathology, Institute of Agricultural Sciences, Banaras Hindu University, Varanasi [29]. Following multiplication of the isolate on sorghum grain, a suspension in water of $10^4$ spores/ml was applied uniformly at the heading stage [30], with the spraying being done in the evening [31]. The field was irrigated the next morning to provide a favourable environment for disease development.

## Scoring for lesion mimic, leaf tip necrosis, SB, glaucous index and latent period

The plants were observed in the experimental plot for the traits Lm, *Ltn*, SBGI and LP. Five randomly tagged plants of each genotype were evaluated for the expression of Lm symptoms. Lm was scored on flag leaf at growth stages (GS) 63, 69 and 77. Leaves showing typical Lm expression were scored with modification of the 1–9 rating scale [13]. Within the scale, the % area denotes the leaf area necrosis, where 1 = no visible specks, 2 = 1–10%, 3 = 21–30%, 4 = 31–40%, 5 = 41–50%, 6 = 51–60%, 7 = 61–70%, 8 = 71–80% and 9 = more than 80%. Flag leaves of the tagged plants of each genotype were evaluated for *Ltn* at GS69. For *Ltn*, leaves were scored in two ways—first, as a presence or absence of *Ltn*; second, when present, scored as the level of its expression on a scale of 1–5, where 1 = no *Ltn*, 2 = 25%, 3 = 50%, 4 = 75% and 5 = necrosis of more than 75% of the flag leaf. Glaucous or waxiness on the 5 tagged plants was recorded visually at the time of flowering on the peduncle and flag leaf sheath on a scale of 1–5. Here, 1 denotes a very low or minimum appearance of waxiness, 2 denotes low waxiness appearance, 3 denotes a comparatively moderate level of waxiness, 4 denotes a high level of waxiness appearance while 5 indicates a maximum level of waxiness.

LP is the period in days between inoculation and spore production and was observed and recorded using the process described by Parlevliet [32]. Five randomly selected flag leaves from each tagged plant were examined with the aid of a 20 × magnifying lens to establish when 50% of the primary lesions were sporulated.

## Disease assessment

Ten randomly tagged plants of each genotype were evaluated for SB severity at three different growth stages, GS63 (beginning to half-completion of anthesis), GS69 (anthesis complete) and GS77 (late milking) using a double-digit scale (DD, 00–99) according to Saari and Prescott [33]. For each score, the disease severity percentage was calculated using the formula:

$$\%severity = (D1/9)(D2/9)100$$

Where,

$$D_1 = \textit{vertical disease progress on the plant}$$
$$D_2 = \textit{the disease severity score on the affected leaves}$$

AUDPC was based on disease severity at GS63, GS69 and GS77 using the percent severity estimates as outlined in [34], given as:

$$AUDPC = \sum_{i=0}^{n-1} [\{(Y_i + Y_{i+1})/2\} \times (t_{i+1} - t_i)]$$

Where,

$$Y_i = \textit{disease severity at time } t_i$$
$$(t_{i+1} - t_i) = \textit{time interval } (days) \textit{ between two disease scores}$$
$$n = \textit{the number of dates at which SB was recorded}$$

## DNA extraction and SNP genotyping

DNA was extracted from 20-day fresh leaves of each line following the CTAB procedure [35] and genotyped at CIMMYT, Mexico using the Illumina iSelect beadchip assay [36] for wheat. To avoid low polymorphic and low-quality SNPs, markers were filtered on the parameter of minor allele frequency $< 0.10$. Thus 13,589 out of a total of 15,737 highly polymorphic SNPs were selected and used for association mapping.

## Phenotypic and population structure analysis

Analysis of Variance (ANOVA) was carried out to determine genotype, year, and genotype × year variances among the traits measured. Correlation analysis was performed to better understand the relationship among the traits. All these analyses were done using SAS 9.3. The population structure (Q) for the WAMI marker panel was determined using the program STRUCTURE v2.3.4 [37]. The number of clusters (K) was predefined as1–10 with a burn-in of 10,000 iterations followed by 10,000 Markov Chain Monte Carlo (MCMC) replicates, passed as initial parameters for running STRUCTURE. The number of subgroups of the population was estimated using 'Structure Harvester' [38], a web-based utility that provides maximum likelihood estimates of the proportion of each sample derived from each of the K populations. The population Q-matrixwas also obtained for further analysis.

## Genome-wide association analysis

TASSEL 5.0 [39] was used for the identification of significant marker-trait associations, based on the Mixed Linear Model (MLM). MLM takes into account both the population structure (Q-matrix generated through STRUCTURE) as well as the ancestral relatedness i.e. kinship matrix (K). TASSEL 5.0 was used to calculate the population kinship matrix by applying a scaled Identity By State (IBS) method. The general mathematical formulation of this mixed linear model can be written in the following form:

$$y = Xa + Qb + Ku + e$$

Where,

$$y = \text{the vector of phenotypes}$$
$$a \text{ and } b = \text{vectors of fixed effects}$$
$$u = \text{the vector of random effects (Kinship matrix)}$$
$$e = \text{the vector of random residuals}$$
$$X = \text{the genotypes of marker}$$
$$Q = \text{the population structure}$$
$$K = \text{kinship matrix}$$

Since the Q-matrix is used as a covariate in the model, it controls the structure and also avoids false positives. MLM is used preferentially because of its efficiency in terms of reducing time complexity [40]; its parameters were left at the default settings when running TASSEL. AP-value ≤0.001 was taken as denoting a significant marker-trait association (MTA) and the $R^2$ value was used to evaluate the magnitude of the QTL effects. For better visualization of results, Manhattan plots were also generated. Linkage distribution among the markers was also calculated.

## Results

### Phenotypic analysis

The results of the ANOVA for the five measured traits of the WAMI panel over three consecutive growing seasons are presented in Table 1. It can be observed that the genotypes exhibit differences at a 1% level of significance for all the traits. The year was also found to be significant for all traits except *Ltn*. Moreover, significant differences were observed for genotype × year (P<0.01) for all the traits. Partitioning of the total sum of squares indicates that the year accounted for more variation than genotype for all the traits except *Ltn*. In addition, AUDPC for SB was negatively and significantly correlated with *Ltn* (-0.302) and LP (-0.529), while positively and significantly with Lm (0.493);*Ltn* is positively correlated with LP (0.313) but negatively with Lm (-0.426) (Table 2).

### Population structure and linkage disequilibrium analysis

From model-based analysis using STRUCTURE (Figs 1 & 2) the optimal K was determined to be 6. Subpopulation I contained 58 (20.1%) genotypes; II, 56 (19.3%); III, 74 (25.6%); IV, 35 (12.1%); V, 48 (16.7%); and VI, 18 (6.2%). Individuals of each population were categorized as pure or admixture types. Genotypes with≥0.8 of member proportions were considered as pure were others were labeled admixtures. Considering this criterion, the composition of the six subpopulations was as follows; I, 2.8% pure and 17.3% admixture; II, 4.8% pure and 14.5%

**Table 1. Analysis of variance for the traits lesion mimic, leaf tip necrosis, latent period, glaucous index and area under disease progress curve.**

| Source | Df | Lm | | *Ltn* | | LP | | GI | | AUDPC | |
|--------|-----|-------------|---------|-------------|---------|-------------|---------|-------------|---------|-------------|---------|
| | | Mean square | F Value | Mean square | F Value | Mean square | F Value | Mean square | F Value | Mean square | F Value |
| Genotype | 288 | 2199.76* | 61.54 | 648.95* | 19.30 | 14.87* | 14.99 | 2.37* | 6.49 | 123267.82* | 99.50 |
| Year | 2 | 1890.15* | 52.88 | 117.87 | 3.51 | 206.35* | 207.86 | 3.63* | 9.92 | 3170105.74* | 2558.98 |
| Replication | 1 | 5.20 | 0.15 | 36.04 | 1.07 | 0.27 | 4.94 | 1.80 | 4.94 | 2902.96 | 2.34 |
| Genotype × year | 576 | 146.95* | 4.11 | 70.65* | 2.10 | 2.02* | 2.06 | 0.75* | 2.06 | 7135.74* | 5.76 |

*Significant at P<0.01.

**Table 2. Pearson correlation coefficients analysis for the five traits studied.**

| Traits | AUDPC | GI | *Ltn* | Lm |
|--------|-------|----|-------|-----|
| **GI** | -0.052 (0.02) | 1 | | |
| ***Ltn*** | -0.302 (< .001) | 0.009 (0.0681) | 1 | |
| **Lm** | 0.493 (< .001) | -0.092 (0.0001) | -0.426 (< .0001) | 1 |
| **LP** | -0.529 (< .001) | 0.063 (0.0072) | 0.313 (< .0001) | -0.465 (< .0001) |

AUDPC = Area under disease progress curve; GI = Glaucousness index; *Ltn* = Leaf tip necrosis; Lm = Lesion mimic; LP = Latent period. P-values are given in parentheses.

admixture; III, 3.1% pure and 22.5% admixture; IV, 1.4% pure and 10.7% admixture; V, 4.2% pure and 12.5% admixture; and VI, 1.4% pure and 4.8% admixture. Fig 2 shows the population structure. To evaluate the population composition, Q-matrix (K = 6) and Kinship matrix were further used as covariates for a GWAS. A total of 13,589 SNPs markers that passed quality filtering were used for mapping. Of these SNPs, 4,967 had loci mapped on A genome, 7,236 on B genome and 1,386 on D genome (S2 Table). A Linkage Disequilibrium (LD) plot based on the association among the markers was generated (Fig 3). LD was estimated from all pairs of SNPs along each chromosome. The average LD $R^2$ was 0.35 for the A sub-genome, 0.37 for B and 0.36 for D. In Fig 3 most of the markers are tightly linked and observed below the diagonalas large areas of red. This denotes that there was restricted space for recombination between the markers, which facilitates association mapping of the five traits, and suggests that a minimum number of markers is required effectively to cover the entire genome. A more detailed distribution of SNPs over chromosomes is presented in S1 Table.

## Genome-wide marker-trait association with SNP-markers

A total of 29 SNPs exhibited significant marker-trait associations with Lm, GI, Ltn, LP and SB AUDPC at P<0.001. They occur across seven different genomic regions (1B, 3B, 3D, 5B, 5A,

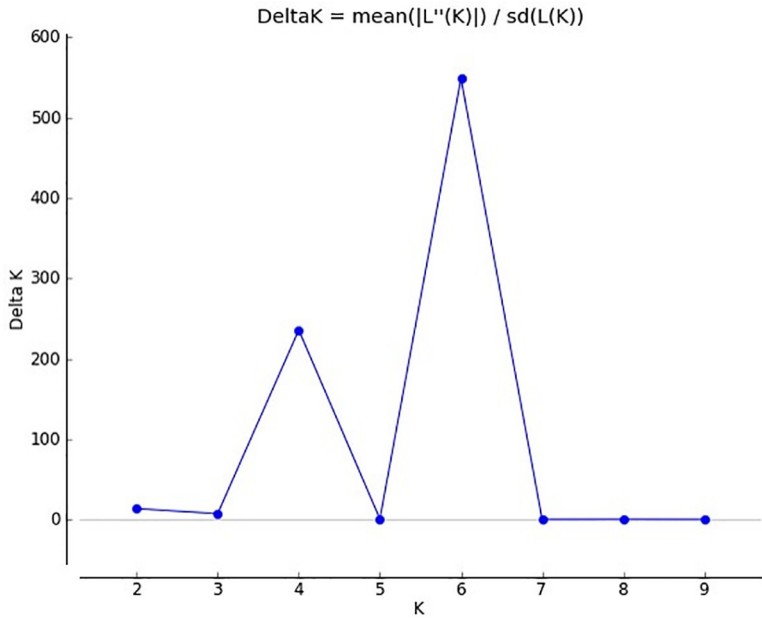

**Fig 1. Population structure showing genetic relationships of 289 wheat lines.** Δ*K* plot, with *K* ranging from 1–10.

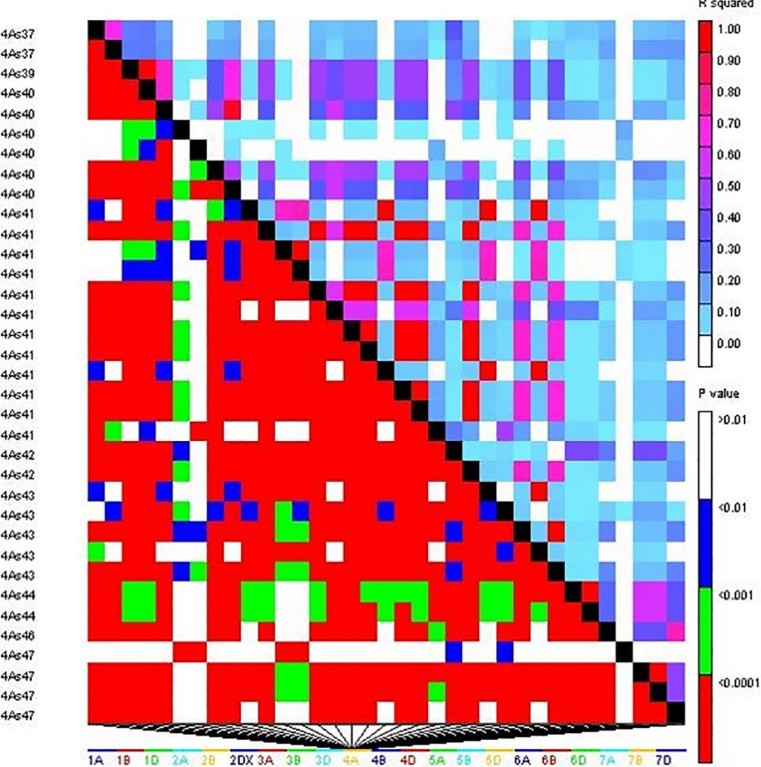

**Fig 2. STRUCTURE analysis used to define genetic relationships among 289 wheat lines.** The existence of six subpopulations was inferred. Plot was generated using the mean of the variation posterior distribution over inferred admixture proportions. The X-axis shows the membership coefficients and Y-axis shows the different genotype entries. A visual vertical separation represents different subpopulations.

6A and 6B). Seven SNPs each were found for Lm and GI, one SNP for *Ltn*, five SNPs for LP and nine SNPs for SB AUDPC. The P- and $R^2$-values, and other details of these marker-trait associations are presented in Table 3. The Manhattan plots for each trait are shown in Fig 4. Individually the SNPs explain 5-8% of the total phenotypic variation.

For lesion mimic a total of seven significant marker trait associations (MTAs) were mapped on chromosome 6A with four situated at 90 cM and the remaining three at 91cM. Each of these markers explained 6–7% of total phenotypic variation. A single SNP marker was significantly associated with *Ltn* and mapped on chromosome 1B at a distance of 115 cM. This SNP explained 6% of total phenotypic variation. Seven SNPs were found to be in significant marker-trait association with GI. One marker was on chromosome 6A at 21 cM. All others

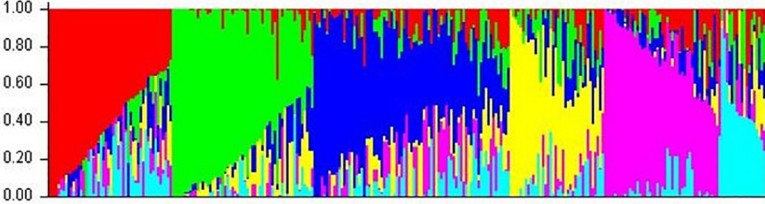

**Fig 3. Linkage disequilibrium plot of significant marker associations.** $R^2$ and *P*-values of pair-wise analyses are indicated by colour in the right-side bars.

**Table 3. List of significant SNPs associated with five different traits—Lm, *Ltn*, GI, LP and AUDPC—detected in the 289 WAMI spring wheat panel.**

| Trait | Marker | Chr. | Pos | *P* value | Marker $R^2$ |
|---|---|---|---|---|---|
| **Lm** | wsnp_CAP11_c1178_684471 | 6A | 90 | 2.39E-05 | 0.0694 |
| | wsnp_Ra_c12086_19452422 | 6A | 91 | 2.88E-05 | 0.0641 |
| | Tdurum_contig69065_319 | 6A | 91 | 3.16E-05 | 0.0665 |
| | Tdurum_contig55363_297 | 6A | 90 | 3.36E-05 | 0.0635 |
| | wsnp_Ku_rep_c102901_89769309 | 6A | 91 | 4.1E-05 | 0.06181 |
| | wsnp_RFL_Contig3136_3092151 | 6A | 90 | 4.37E-05 | 0.06497 |
| | Tdurum_contig29974_90 | 6A | 90 | 5.67E-05 | 0.06444 |
| ***Ltn*** | Ex_c25733_348 | 1B | 115 | 7.69E-04 | 0.05931 |
| **GI** | BobWhite_c3714_659 | 6A | 21 | 4.55E-06 | 0.07696 |
| | CAP7_c524_326 | 6B | 118 | 5.85E-06 | 0.07547 |
| | Kukri_rep_c79491_139 | 6B | 118 | 1.14E-05 | 0.07075 |
| | TA001682-1583 | 6B | 119 | 3.8E-05 | 0.06106 |
| | RAC875_c17011_373 | 6B | 122 | 4.17E-05 | 0.06214 |
| | RAC875_c21938_1408 | 6B | 120 | 4.87E-05 | 0.06042 |
| | TA002907-0816 | 6B | 122 | 8.65E-05 | 0.05608 |
| **LP** | RAC875_c4389_1344 | 3B | 11 | 8.83E-06 | 0.07531 |
| | RAC875_c4389_1412 | 3B | 32 | 1.02E-05 | 0.07691 |
| | tplb0043c20_1046 | 3B | 26 | 3.38E-05 | 0.06421 |
| | tplb0043c20_1046–1 | 3D | 18 | 3.38E-05 | 0.06421 |
| | GENE-1851_76 | 3B | 26 | 8.36E-05 | 0.05819 |
| **AUDPC** | wsnp_Ku_c40334_48581010 | 5B | 90 | 3.9E-05 | 0.06109 |
| | BobWhite_c48435_165 | 5B | 90 | 6.53E-05 | 0.05752 |
| | Tdurum_contig12066_126 | 5A | 83 | 7.01E-05 | 0.05716 |
| | Tdurum_contig12066_247 | 5A | 83 | 7.01E-05 | 0.05716 |
| | Tdurum_contig12066_126–1 | 5B | 90 | 7.01E-05 | 0.05716 |
| | Tdurum_contig12066_247–1 | 5B | 90 | 7.01E-05 | 0.05716 |
| | tplb0027f13_1493 | 5B | 90 | 7.85E-05 | 0.05702 |
| | tplb0027f13_1346 | 5A | 83 | 8.01E-05 | 0.05616 |
| | tplb0027f13_1346–1 | 5B | 90 | 8.01E-05 | 0.05616 |

were on chromosome 6B. Two of these were each at 118 cM, another two were at 119 and 120 cM, while the remaining two were mapped at a distance of 122 cM. The phenotypic variations explained by these loci were in the range 5–8%. For LP, five significant SNPs were identified on chromosome 3B and 3D. Two mapped together at 26cM on chromosome 3B which suggests that a single QTL for LP may link to these SNPs. Another two occurred on chromosome 3B at 11 cM and 32 cM respectively, while the remaining marker was on chromosome 3D at 18 cM. Each explained 6–8% of total genetic variation. A total of nine SNPs on chromosomes 5A and 5B were found to be in significant marker-trait associations for AUDPC (Table 3). Six of these SNPs were on chromosome 5B, each at 90 cM; the other three clustered on 5A at 83 cM. It is possible that two QTLs for SB resistance may link to these SNPs. The phenotypic variation explained by each individual locus was around 6%.

## Identification of putative candidate genes and their annotation

To analyze and further annotate significant MTAs, we located them on a reference wheat genome (RefSeq v1.0). Due to the large size of the wheat genome, extended 250 kb genomic regions either side of significantly associated SNPs were analyzed to identify putative genes. A

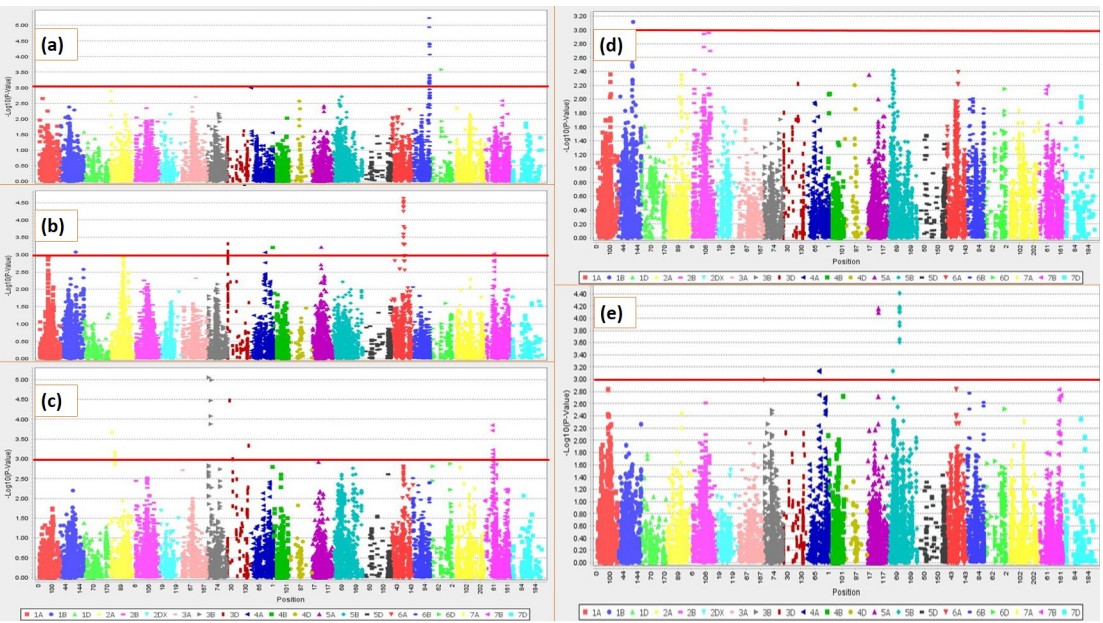

**Fig 4.** Manhattan plots for different traits under study: a) GI b) LM c) LP d) *Ltn* e) AUDPC. The threshold line at P = 0.001 has been drawn to highlight significant markers.

total of 27 candidate genes were annotated by function (Table 4). The MTAs that could not be annotated were Kukri_rep_c79491_139, associated with GI, and Tdurum_contig_12066_126 associated with AUDPC.

For trait Lm, markers mapped within a very narrow interval of 0.09 Mbpon chromosome 6A, in the region from 56.34 Mbp (wsnp_Ku_rep_c102901_89769309 and wsnp_CAP11_c1178_684471) to 56.43 Mbp (Tdurum_contig55363_297). This interval contains seven genes, of which five have annotations with high confidence (Table 4). Few of the candidates genes having more than one SNPs at same chromosome position like TraesC-S6A02G33100 (wsnp_Ra_c12086_19452422, wsnp_Ku_rep_c102901_89769309) and TraesC-S6A02G331000(wsnp_CAP11_c1178_684471, wsnp_RFL_Contig3136_3092151) at 91 and 90 cM respectively. These genes encode for various classes of proteins and enzymes including a WD40-repeat-containing domain superfamily member (which regulates a plant-specific developmental event to control cell cycle) and MAG2-interacting protein (which acts as a precursor for the accumulation in dry seeds of the two major storage proteins albumin 2S and globulin 12S. The single marker identified for *Ltn* was located on chromosome 1B and belongs to the protein kinase-like superfamily, which is common to both serine/threonine and tyrosine protein kinases and has a catalytic domain that contains anucleotide-binding site (NBS), playing a critical role in disease resistance.

Similarly, for trait LP, most markers mapped on to chromosome 3B in a 0.21 Mbp interval at 0.90–1.11 Mbp. Likewise for lesion mimic here alsowe found two SNPs falling under same candidate gene viz. TraesCS3B02G025200 (tplb0043c20_1046, GENE-1851_76). Their underlying identified genes were found to relate to the Cytochrome P450 superfamily and Fructose-bisphosphate aldolase, which are involved in specific mechanisms like stress and defense response, energy and metabolism.

For GI, six of the seven MTAs mapped on to chromosome 6B. The identified genes related to Ubiquitin-conjugating enzyme/RWD-like proteins (with functions involved with plant innate immunity), Synaptotagmin-like mitochondrial-lipid-binding protein (that acts as

**Table 4. Detailed annotation (i.e. underlying genes, their functions and GO terms) of identified markers for different traits related to spot blotch.**

| Markers | Chromosome | Gene accession | Gene Descriptor | GO Terms |
|---|---|---|---|---|
| wsnp_CAP11_c1178_684471 | 6A | TraesCS6A02G331000 | Mitochondrial carrier domain superfamily | GO:0016021 |
| wsnp_Ra_c12086_19452422 | 6A | TraesCS6A02G333100 | WD40-repeat-containing domain superfamily, MAG2-interacting protein | GO:0005737, GO:0006888, GO:0006890, GO:0032527 |
| Tdurum_contig69065_319 | 6A | TraesCS6A02G333400 | Glycoside hydrolase, family 5 | |
| Tdurum_contig55363_297 | 6A | TraesCS6A02G332000 | Mitochondrial carrier domain superfamily | GO:0016021 |
| wsnp_Ku_rep_c102901_89769309 | 6A | TraesCS6A02G333100 | WD40-repeat-containing domain superfamily, MAG2-interacting protein | GO:0005737, GO:0006888, GO:0006890, GO:0032527 |
| wsnp_RFL_Contig3136_3092151 | 6A | TraesCS6A02G331000 | Mitochondrial carrier domain superfamily | GO:0005618, GO:0005774, GO:0005794, GO:0031305, GO:0005315, GO:0009651, GO:0035435 |
| Tdurum_contig29974_90 | 6A | TraesCS6A02G331500 | Cyclin-like superfamily | |
| Ex_c25733_348 | 1B | TraesCS1B02G419400 | Protein kinase-like domain superfamily | |
| BobWhite_c3714_659 | 6A | TraesCS6A02G016200 | Ubiquitin-conjugating enzyme/RWD-like | |
| CAP7_c524_326 | 6B | TraesCS6B02G463000 | Synaptotagmin-like mitochondrial-lipid-binding domain | GO:0016021, GO:0008289, GO:0006869 |
| Kukri_rep_c79491_139 | 6B | NA | | |
| TA001682-1583 | 6B | TraesCS6B02G462300 | Haem peroxidase superfamily | |
| RAC875_c17011_373 | 6B | TraesCS6B02G472900 | Lunapark family | GO:0016021, GO:0071786 |
| RAC875_c21938_1408 | 6B | TraesCS6B02G465300 | AP-5 complex subunit beta-1 | GO:0005623, GO:0016021, GO:0016197 |
| TA002907-0816 | 6B | TraesCS6B02G471900 | Domain unknown function DUF295 | |
| RAC875_c4389_1344 | 3B | TraesCS3B02G025600 | Cytochrome P450 superfamily | GO:0004497, GO:0005506, GO:0016705, GO:0020037, GO:0055114 |
| RAC875_c4389_1412 | 3B | TraesCS3B02G025600 | Cytochrome P450 superfamily | GO:0004497, GO:0005506, GO:0016705, GO:0020037, GO:0055114 |
| tplb0043c20_1046 | 3B | TraesCS3B02G025200 | Fructose-bisphosphate aldolase, class-I | GO:0004332, GO:0006096, EC:4.1.2.13 |
| tplb0043c20_1046–1 | 3D | TraesCS3D02G026400 | Fructose-bisphosphate aldolase | GO:0004332, GO:0006096, EC:4.1.2.13 |
| GENE-1851_76 | 3B | TraesCS3B02G025200 | Aldolase-type TIM barrel | GO:0005829, GO:0016021, GO:0004332, GO:0006096, GO:0030388, EC:4.1.2.13 |
| wsnp_Ku_c40334_48581010 | 5B | TraesCS5B02G368600 | S-acyltransferase | GO:0016021, GO:0019706, EC:2.3.1.225 |
| BobWhite_c48435_165 | 5B | TraesCS5B02G368500 | Potassium transporter | GO:0016021, GO:0015079, GO:0071805 |
| Tdurum_contig12066_126 | 5A | TraesCS5A02G366100 | Potassium transporter | GO:0016021, GO:0015079, GO:0071805 |
| Tdurum_contig12066_126–1 | 5B | NA | | |
| Tdurum_contig12066_247 | 5A | TraesCS5A02G366100 | Potassium transporter | |
| Tdurum_contig12066_247–1 | 5B | TraesCS5B02G368500 | Potassium transporter | |
| tplb0027f13_1493 | 5B | TraesCS5B02G368500 | Potassium transporter | GO:0016021, GO:0015079, GO:0071805 |
| tplb0027f13_1346 | 5A | TraesCS5A02G366100 | Potassium transporter | GO:0016021, GO:0015079, GO:0071805 |
| tplb0027f13_1346–1 | 5B | TraesCS5B02G368500 | Potassium transporter | GO:0016021, GO:0015079, GO:0071805 |

molecular hubs for the exchange of small molecules such as lipids, and of signals, such as calcium ions), haem peroxidase superfamily members (which act as an electron acceptor to catalyze several oxidative reactions), the Lunapark family, and AP-5 complex subunit beta-1 (i.e. floral organ development and plant reproduction).

A total of 9 SNPs were found to be in significant association to AUDPC of which 6 SNP mapped on chromosome 5B, four SNPs (BobWhite_c48435_165, tplb0027f13_1493, Tdurum_contig12066_247–1 and tplb0027f13_1346–1) represents same candidate gene i.e. TraesCS5B02G368500, whereas SNP wsnp_Ku_c40334_48581010 represent candidates gene involved in S-acyltransferase (i.e. TraesCS5B02G368600) and remaining one

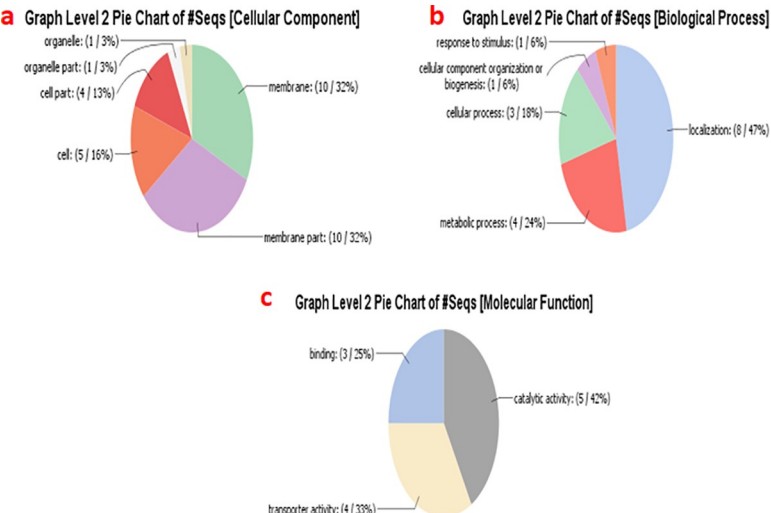

**Fig 5.** Graphical representation of categories of Gene Ontology (GO) terms of identified MTA related to Spot blotch: a) Cellular Component b) Biological Process c) Molecular Function.

(Tdurum_contig12066_126–1) could not be annotated (Table 4). Remaining 3 SNPs for AUDPC mapped on chromosome 5A (at 83 cM i.e. Tdurum_contig12066_126, Tdurum_contig12066_247 and tplb0027f13_1346) represent the same candidate gene i.e. TraesCS5A02G366100. Seven of the nine MTAs identified were found to be involved in potassium transport which has a crucial role in plant responses to, and tolerance of, abiotic stresses. Detailed annotations (i.e. cellular component, biological process and molecular function) are shown in Fig 5.

## Discussion

Lm in wheat is governed by four recessive genes—*lm*, *lm1*, *lm2* [13] and *lm3* [41]. Since they are known to demonstrate a protective effect against biotrophic pathogens [12,27,41], Lm genes are being introduced in wheat genotypes to provide resistance against these pathogens. However, these genes express their symptoms as cell death in the leaf tissues of the host genotypes. If this expression is severe, there may be a negative effect due to a reduced photosynthetic area. In addition, the dead tissue might be a nutrient source for various hemibiotrophs and necrotrophs. Therefore it is necessary to establish the effects on Lm plants of diseases like SB, which is one of the major concerns for the Eastern Gangetic Plains (EGP) region of South Asia covering the >10 Mha wheat belt of India, Nepal and Bangladesh [42–44].

Analysis of variance indicated the presence in the WAMI spring wheat panel of significant variability for Lm and the other traits investigated. Lm was positively correlated with AUDPC which indicates that it promotes SB severity, confirming previous reports [45]. Lm appears to enhance the impact of SB due to the hemibiotrophic nature of *B.sorokiniana*, which germinates on living cells but multiplies on dead cells. Since Lm contributes to SB progress, reduced Lm expression can control the necrotrophic action of the pathogen. The results of this study demonstrate that higher LP is associated with lower levels of SB and Lm. Higher levels of GI can also restrict the progress of SB. Therefore, for the development of SB-resistant genotypes, a combination of *Ltn* and GI with higher LP may be utilized, which can reduce Lm gene expression as well as SB disease.

An inhibitory effect of *Ltn* on Lm was observed in this study. The association of LP with *Ltn* was positive, which indicates that resistance against SB is enhanced by the presence of *Ltn* [17]. *Ltn* can thus be recommended as a phenotypic marker for the selection of SB resistant

genotypes. Recently, Singh et al. [46] proposed that stacking *Lr34*(*Sb1*), *Lr46*, and *vrn-A1* (which are prevalent in the CIMMYT gene pool) with additional SB resistance QTL can lead to a high level of SB resistance. *Ltn* in wheat is associated with *Lr34* [47] and it can provide resistance against rust as well as SB [17,47]. Glaucous index also showed a positive association with *Ltn* indicating the cumulative positive effect of *Lr34*and increased waxiness on SB resistance. For this reason, genotypes with well-expressed GI or waxiness also appear to inhibit the expression of Lm. Similarly, LP displayed a negative association with Lm, and longer LP contributed to enhanced levels of resistance against SB.

In this study, GWAS was performed using 90 k Illumina SNPs chip markers to establish the genotypic relationships of five traits—Lm, *Ltn*, LP, GI, and AUDPC for SB. To validate the broader applicability of SNPs and GWAS, we also verified the resistances that were previously detected using DArT markers [48–50]. Nine QTLs relating to AUDPC were identified on chromosomes 5A and 5B for improved resistance against SB.

In the association study of Adhikari et al. [49], genomic regions associated with SB resistance led to the identification of nine SNPs on chromosomes 1B, 5A, 5B, 6B, and 7B. The study used 528diverse spring wheat genotypes that were phenotyped for SB and genotyped utilizing a 9K SNP wheat chip [51]. Ahirwar et al. [52] reported 14 SNPs on chromosomes 1B, 5B, 6A and 6B. We detected MTAs that corresponded to nine previously-reported loci on two chromosomes [47,53,54] from biparental QTL analyses. The present study identified QTLs for SB resistance that were reported previously at similar positions which confirms the robust inheritance of QTLs associated with the SNPs mapped here.

This work is the first to study Lm and its association with LP, *Ltn*, GI, and AUDPC for SB. Novel SNPs for these traits were firmly identified. The genotypic relationships among these traits need to be further studied for improvement of the wheat research program. The reported QTLs from this study should provide a foundation for further research in this area.

## Conclusions

Twenty nine significant marker-trait associations were identified in the present investigation. We found seven markers closely associated with Lm, all on chromosome 6A, one for *Ltn* on 1B, and seven for GI across 6A and 6B. Five markers for LP were mapped on chromosomes 3B and 3D while nine SNPs on 5A and 5B were associated with AUDPC for SB.

A notable phenotypic and molecular variation was observed in the WAMI panel, which confirms the diverse genetic background of the WAMI germplasm. The genotype analysis showed significant positive correlations between Lm and AUDPC, *Ltn* and LP, and between LP and GI, whereas Lm was significantly negatively associated with *Ltn*, LP, and GI.

This study established for the first time an association of markers for Lm, *Ltn*, LP and GI, and QTLs mapped through GWAS. Our data revealed that most of the SNPs were present on the A and B genomes of wheat. These identified SNP markers linked to different QTLs will be useful in breeding for Lm and for SB resistance in wheat. The study also establishes a clear association between Lm and *Ltn* with AUDPC for SB, GI and LP. Based on a positive association of Lm and AUDPC, the pattern of appearance of HR, and necrosis and lesion formation by Lm, it is evident that the structure, expression, function and pathways of Lm genes can provide useful information. Utilizing this information to better understand the nature of SB pathogens is will be critical in the development and selection of resistance cultivars.

## Supporting information

**S1 Table. Marker distribution among the population studied.**
(DOCX)

**S2 Table. Distribution of 13589 highly polymorphic SNPS throughout the wheat genome.** (DOCX)

**S3 Table. List of lines in the spring wheat association mapping (WAMI) evaluated over three years in BHU, Varanasi.** (DOCX)

## Acknowledgments

The authors gratefully acknowledge Matthew P. Reynolds, CIMMYT, Mexico, for providing WAMI population and the open-access molecular data used in this study.

## Author Contributions

**Conceptualization:** Vinod Kumar Mishra, Ravindra Nath Kharwar, Ramesh Chand, Arun Kumar Joshi.

**Data curation:** Vinod Kumar Mishra, Neeraj Budhlakoti, Ram Narayan Ahirwar, Sundeep Kumar, Ramesh Chand.

**Formal analysis:** Neeraj Budhlakoti, Ram Narayan Ahirwar, Dwijesh Chandra Mishra, Sundeep Kumar.

**Investigation:** Shweta Singh, Vinod Kumar Mishra, Ravindra Nath Kharwar.

**Methodology:** Shweta Singh, Vinod Kumar Mishra, Ravindra Nath Kharwar, Ramesh Chand, Uttam Kumar, Arun Kumar Joshi.

**Project administration:** Vinod Kumar Mishra.

**Resources:** Vinod Kumar Mishra, Ramesh Chand, Uttam Kumar, Arun Kumar Joshi.

**Software:** Neeraj Budhlakoti, Dwijesh Chandra Mishra, Sundeep Kumar.

**Supervision:** Vinod Kumar Mishra, Arun Kumar Joshi.

**Validation:** Shweta Singh, Vinod Kumar Mishra, Ram Narayan Ahirwar.

**Visualization:** Shweta Singh, Ram Narayan Ahirwar, Dwijesh Chandra Mishra, Sundeep Kumar.

**Writing – original draft:** Shweta Singh.

**Writing – review & editing:** Vinod Kumar Mishra, Ravindra Nath Kharwar, Neeraj Budhlakoti, Ram Narayan Ahirwar, Ramesh Chand, Uttam Kumar, Suneel Kumar, Arun Kumar Joshi.

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
