## [Decision Letter · Decision Letter 0]

10 Aug 2019

PONE-D-19-19629

Genetic Characterization for Lesion mimic and other Traits in Relation to Spot blotch Resistance in Spring Wheat

PLOS ONE

Dear Dr. Mishra,

Thank you for submitting your manuscript to PLOS ONE. After careful consideration, we feel that it has merit but does not fully meet PLOS ONE’s publication criteria as it currently stands. Therefore, we invite you to submit a revised version of the manuscript that addresses the points raised during the review process.

ACADEMIC EDITOR: 

**You will see that both the reviewers made critical recommendations, but an important consideration is whether the study describes a technically sound research and made significant advances in the relevant area. However, as reviewers pointed out, I think that this manuscript needs additional works. Therefore, I encourage the authors to perform the revision as per the reviewer’s suggestions and if these were meticulously performed, then I am sure that the MS could be reconsidered on a later date **

We would appreciate receiving your revised manuscript by Sep 24 2019 11:59PM. To enhance the reproducibility of your results, we recommend that if applicable you deposit your laboratory protocols in protocols.io, where a protocol can be assigned its own identifier (DOI) such that it can be cited independently in the future. For instructions see: http://journals.plos.org/plosone/s/submission-guidelines#loc-laboratory-protocols

We look forward to receiving your revised manuscript.

Kind regards,

Manoj Prasad, PhD

Academic Editor

PLOS ONE

Journal Requirements:

1. In your Methods or in a Supporting Information file, please include a full list of all the wheat accessions included in your study.

Reviewers' comments:

Reviewer's Responses to Questions

**Comments to the Author**

1. Is the manuscript technically sound, and do the data support the conclusions?

Reviewer #1: Partly

Reviewer #2: Partly

2. Has the statistical analysis been performed appropriately and rigorously? 

Reviewer #1: Yes

Reviewer #2: No

3. Have the authors made all data underlying the findings in their manuscript fully available?

Reviewer #1: Yes

Reviewer #2: No

4. Is the manuscript presented in an intelligible fashion and written in standard English?

Reviewer #1: No

Reviewer #2: No

5. Review Comments to the Author

Reviewer #1: Spot blotch is an important wheat disease in tropical and sub-tropical regions, especially India, China, Mexicao, etc. Identifying genetic loci responsible for Spot blotch resistance are important for wheat breeding and also for following genomic analysis. The manuscript performed genome wide association analysis of spot blotch (AUDPC) and lesion mimic (Lm) as well as leaf tip necrosis (Ltn), glaucousness index (GI) and latent period (LP) using a 289 GWAS panel and SNP assay. Positive association between AUDPC of spot blotch and Lm was found and negative correlation between Lm and GI, Ltn and LP were observed. SNP loci associated with Lm, AUDPC of spot blotch, were also identified. These information are valuable for wheat breeding and further fine gene mapping and cloning.

However, this manucript didn't conduct further analysis of their findings using currently available wheat genomic sequences resources which makes the current manusciprt less informative for the community. The authors should perform further analysis of the identified SNP loci, for example Lm on 6A (90-91 cM), AUDPC on 5A and 5B, etc.) in putative gene annotation and breeder friendly markers (like KASP) markers development and validation.

In addition, the authors found the AUDPC of spot blotch associated SNPs on 5A and 5B are deirved from the same sets of probes. This should be explained further to clearify if these two locus on 5A and 5B are orthologous? or just missclassification of SNP signals of the same probe.

Reviewer #2: The manuscript presents results aiming at analysing relationships between several wheat traits (lesion mimic symptoms, leaf tip necrosis symptom, glaucousness index, latent period, and resistance to spot blotch), and identifying markers associated to those traits. Several issues prevent publication at this stage, they are indicated below.

* Materials and methods need to be described with more details. For example, the rationale of the choice of varieties used needs to be described.

* Because the description of disease assessment is not sufficiently described, one cannot assess if the analyses conducted from these assessments are correct. More specifically, the "2-digit" scale needs to be explicitly described. It may be that this scale mixes two very different assessments, and that the analysis conducted is not correct. This potential issue needs to be thoroughly addressed.

* The Introduction, Results, and Discussion sections need to be improved, see detailed comments below.

* The text needs to be edited. I provide examples for the beginning of the manuscript, starting at the Introduction, in order to illustrate the editing work required for the whole manuscript: L51 "Lesion mimic are necrotic symptoms…" : should be "Lesion mimic is …"; L52: HR needs to be spelled out the first time it is used; L53-4; there should be comas between te different items; L54: a space is missing before "Lm". Lm needs to be spelled out the first time it is used; L56: "restricts", not "restrict"; L57: "in the case" not "in case"

More detailed comments are given below.

L53-4: are these all examples with biotroph pathogens? This should be stated early.

L59: is it specifically about Lm plants?

L60: what is ltn?

L63: the rust(s) involved need to be indicated here

L65: spell out the latin name in first instance

L65-67: this is too vague and needs more details.

L70: this needs rephrasing

L70-72: this needs rephrasing as the logic is not sound; do QTL refer to QTL for lm or for resistance to spot blotch?

L73: what is LP? If this is latency period, why choose this "trait" and not for example infection efficiency?

Introduction: the introduction needs to be re-worked so that the reader can understand why this study was undertaken. For example, what is known or what can be hypothesized from the literature, about relationships between Lm, Ltn, and resistance to spot blotch. For a example, it is not indicated if spot blotch is considered as a necrotroph or biotroph, and this is needed. Actually, moving the first paragraph of Discussion in the Introduction, and reorganizing this section, could be a way to improve the Introduction.

L80-3: why were these varieties chosen, on what basis?

L101: the title needs to be changed

L102-3: the two first sentences seem contradictory; was Lm assessed once or several times?

L103: what was the sampling design for Lm symptoms assessment? How many flag leaves per elementary plot?

L105-6: what is the unit: % of what?

L106: what was the sampling design?

L108: what is the unit: % of what?

L109: what is GI? This is not mentioned in the introduction. Why is it interesting to measure this trait?

L110: on what basis was this assessment made? Is there a diagram? Or is there an operational description for "most waxiness expression"?

L12: the method used to measure LP is needs to be described, as well as the sampling design.

L116-8: the computation of "percent disease severity" needs to be clearly described here.

Tables 1 and 2: what are the values given in the tables?

L179-80: Figure 2 should be described in more details.

Figure 3: more details are needed as footnotes of the figure; X and y axes need to be described.

Table 3: R2 values seem very small; can this translate into meaningful effects of phenotypes?

L263 This should appear in the Introduction as well

L265-6: this statement needs to be pondered with the other positive effects of Lm versus other wheat diseases

L267: higher LP is not "providing protection", but is a consequence of other mechanisms which increase LP

L268-70: this is an hypothesis, which is not correct for LP: high LP is a consequence of other mechanisms, not a mechanism which will per se reduce the expression of Lm gene

L272: this is an hypothesis; correlation does not indicate any underlying mechanisms, but indicates only association

L284-307: a discussion on low R2 values is needed here.

6. PLOS authors have the option to publish the peer review history of their article (what does this mean?). If published, this will include your full peer review and any attached files.

Reviewer #1: Yes: Zhiyong Liu

Reviewer #2: No

---

## [Author Response · Author response to Decision Letter 0]

11 Oct 2019

ONE-D-19-19629

Genetic Characterization for Lesion mimic and other Traits in Relation to Spot blotch Resistance in Spring Wheat

PLOS ONE

Reviewers' comments:

Reviewer's Responses to Questions

Comments to the Author

1. Is the manuscript technically sound, and do the data support the conclusions?

Reviewer #1: Partly

Reviewer #2: Partly

2. Has the statistical analysis been performed appropriately and rigorously? 

Reviewer #1: Yes

Reviewer #2: No

3. Have the authors made all data underlying the findings in their manuscript fully available?

Reviewer #1: Yes

Reviewer #2: No

4. Is the manuscript presented in an intelligible fashion and written in standard English?

Reviewer #1: No

Reviewer #2: No

5. Review Comments to the Author

Reviewer #1: Spot blotch is an important wheat disease in tropical and sub-tropical regions, especially India, China, Mexicao, etc. Identifying genetic loci responsible for Spot blotch resistance are important for wheat breeding and also for following genomic analysis. The manuscript performed genome wide association analysis of spot blotch (AUDPC) and lesion mimic (Lm) as well as leaf tip necrosis (Ltn), glaucousness index (GI) and latent period (LP) using a 289 GWAS panel and SNP assay. Positive association between AUDPC of spot blotch and Lm was found and negative correlation between Lm and GI, Ltn and LP were observed. SNP loci associated with Lm, AUDPC of spot blotch, were also identified. These information are valuable for wheat breeding and further fine gene mapping and cloning.

However, this manucript didn't conduct further analysis of their findings using currently available wheat genomic sequences resources which makes the current manusciprt less informative for the community. The authors should perform further analysis of the identified SNP loci, for example Lm on 6A (90-91 cM), AUDPC on 5A and 5B, etc.) in putative gene annotation and breeder friendly markers (like KASP) markers development and validation.

In addition, the authors found the AUDPC of spot blotch associated SNPs on 5A and 5B are derived from the same sets of probes. This should be explained further to clearify if these two locus on 5A and 5B are orthologous? or just missclassification of SNP signals of the same probe.

Reply: As suggested we have tried to further analysis using currently available wheat genomic resources. We have identified candidate genes for significant marker trait association and carried out their detailed annotation and gene ontology. 

In addition we also addressed your valuable concern (i.e. whether AUDPC of spot blotch associated SNPs on 5A and 5B are derived from the same sets of probes?). We have gone through it carefully; it was observed that there is no orthologous relation between mentioned locus on 5A and 5B. It may be just a miss classification of probe. 

Reviewer #2: The manuscript presents results aiming at analysing relationships between several wheat traits (lesion mimic symptoms, leaf tip necrosis symptom, glaucousness index, latent period, and resistance to spot blotch), and identifying markers associated to those traits. Several issues prevent publication at this stage, they are indicated below.

1) Materials and methods need to be described with more details. For example, the rationale of the choice of varieties used needs to be described.

Reply: The studied population named WAMI germplasm, which is a set of 289 diverse wheat population, provided by Cimmyt, Mexico. Details of the lines are open access 37.

2) Because the description of disease assessment is not sufficiently described, one cannot assess if the analyses conducted from these assessments are correct. More specifically, the "2-digit" scale needs to be explicitly described. It may be that this scale mixes two very different assessments, and that the analysis conducted is not correct. This potential issue needs to be thoroughly addressed.

Reply: I have added detailed in the concerned para of material methods. It is based upon [34] i.e., Saari and Prescott 1975 

3) The Introduction, Results, and Discussion sections need to be improved, see detailed comments below.

Reply: Thank you. All the correction incorporated. 

 The text needs to be edited. I provide examples for the beginning of the manuscript, starting at the Introduction, in order to illustrate the editing work required for the whole manuscript: L51 "Lesion mimic are necrotic symptoms…" : should be "Lesion mimic is …"; L52: HR needs to be spelled out the first time it is used; L53-4; there should be comas between te different items; L54: a space is missing before "Lm". Lm needs to be spelled out the first time it is used; L56: "restricts", not "restrict"; L57: "in the case" not "in case"

More detailed comments are given below.

L53-4: are these all examples with biotroph pathogens? This should be stated early.

Reply:These all are examples of lesion mimic presence and expression in different plants without presence of any pathogen.

L59: is it specifically about Lm plants?

Reply: Yes, it is specifically about Lm plants which shown resistance against biotrophs but may facilitate hemibiotrophs and necrotrophs development.

L60: what is ltn?

Reply: Described in concern para. Ltn is denoted for Leaf tip necrosis which is an adult plant resistance character provides resistance against rust and spot blotch.

L63: the rust(s) involved need to be indicated here

Reply: Rust races like, leaf rust (Puccinia recondite sp. tritici and stripe rust (Puccinia striiformis)

L65: Spell out the latin name in first instance.

Reply: Added as Triticum aestivum L.

L65-67: this is too vague and needs more details.

Reply: Agreed. Explained in detail.

L70: this needs rephrasing.

Reply: Agreed. Rephrased

L70-72: this needs rephrasing as the logic is not sound; do QTL refer to QTL for lm or for resistance to spot blotch?

Reply: Here QTLs refers QTLs associated with resistance against Fusarium head blight and stem rust in wheat.

L73: what is LP? If this is latency period, why choose this "trait" and not for example infection efficiency?

Reply: Latency period was chosen after assumption of positive association of Lm and spot blotch. it was hypothesized that the varieties with longer LP and higher GI are resistance against spot blotch, can also restrict the HR Production of Lm as both HR may follow same pathway in host and can check with same mechanism which may be impart by genes responsible for longer LP and higher GI. It may be a selection criteria for varieties less or no Lm expression.

4: Introduction: the introduction needs to be re-worked so that the reader can understand why this study was undertaken. For example, what is known or what can be hypothesized from the literature, about relationships between Lm, Ltn, and resistance to spot blotch. For a example, it is not indicated if spot blotch is considered as a necrotroph or biotroph, and this is needed. Actually, moving the first paragraph of Discussion in the Introduction, and reorganizing this section, could be a way to improve the Introduction.

Reply: Agreed. Explained in introduction.

L80-3: why were these varieties chosen, on what basis?

Reply: WAMI population is formed by CIMMYT, Mexico consisted of elite genetically diverse germplasm.

L101: the title needs to be changed

Reply: Agreed. Title changed.

L102-3: the two first sentences seem contradictory; was Lm assessed once or several times?

Reply: Field was observed continuously after germination. Once Lm was expressed it was scored for three times at three different growth stages.

L103: what was the sampling design for Lm symptoms assessment? How many flag leaves per elementary plot?

Reply: Alpha lattice design, it was scored on flag leaf at the growth stage (GS) 69, 77 and 85

L105-6: what is the unit: % of what?

Reply: Here % area denoted the leaf area necrosis after HR of Ltn expression.

L106: what was the sampling design?

Reply: Alpha lattice design

L108: what is the unit: % of what?

Reply: Here % area denoted the leaf area necrosis after HR of Lm expression.

L109: what is GI? This is not mentioned in the introduction. Why is it interesting to measure this trait?

Reply: Explained in introduction. Glaucousness or waxiness on plants generally expressed at the time of flowering on the peduncle and flag leaf sheath.This trait restricts HR expansion of spot blotch, on that basis it was hypothesized that it can also check the expansion of HR of Lm and made it a trait of interest to study .

L110: on what basis was this assessment made? Is there a diagram? Or is there an operational description for "most waxiness expression"?

Reply: This sentence is creating confusion. We have just modified it.

L12: the method used to measure LP is needs to be described, as well as the sampling design.

Reply: Agreed, described in concern para as LP, the period between inoculations to spore production (in terms of days), and was recorded on five tagged plants in each replication.

L116-8: the computation of "percent disease severity" needs to be clearly described here.

Reply: have added detailed in the concerned para of material methods and hope now it would clarify the method of calculating AUDPC.

Tables 1 and 2: what are the values given in the tables?

Reply: detailed added in table.

L179-80: Figure 2 should be described in more details.

Reply: Described.

Figure 3: more details are needed as footnotes of the figure; X and y axes need to be described.

Reply: Detail is provided.

Table 3: R2 values seem very small; can this translate into meaningful effects of phenotypes?

Reply: R2 is low but it does not necessarily mean that will not translate into meaningful effects of phenotype. There are many studies reported earlier where R2 is low but still contributing to phenotypes. Details justification regarding this has been provided below.

L263 This should appear in the Introduction as well

Reply: Agreed, added in introduction.

L265-6: this statement needs to be pondered with the other positive effects of Lm versus other wheat diseases.

Reply: Most of the studies on Lm and pathogens are based on biotroph pathogens and reports are about its negative association with concern pathogen.

L267: higher LP is not "providing protection", but is a consequence of other mechanisms which increase LP.

Reply: here we stated that higher LP in association with GI and Ltn found protective against the disease and Lm expansion. We found that genotypes with higher LP in association with Ltn and good waxiness impart negative effect on Lm and spot blotch.

L268-70: this is an hypothesis, which is not correct for LP: high LP is a consequence of other mechanisms, not a mechanism which will per se reduce the expression of Lm gene.

Reply: Here, we concluded after our study on Lm and LP, our data showed that genotypes with higher LP restricted expansion of HR of Lm.

L272: this is an hypothesis; correlation does not indicate any underlying mechanisms, but indicates only association.

Reply: Any association or correlation (if seems important), which is consistence in expression in natural condition could be an important way or parameter to measure the associated characters or to sort out the associated problem so can be a selection criteria in plant breeding and genetics. The underlying mechanism with those associated characters needs to be work out. Here this study was based on three years of trial in two replication, we tried to come out with these conclusion based on our result we got.

L284-307: a discussion on low R2 values is needed here.

Reply: Although R2 is very low for the identified markers but p-value for all the markers are significant enough (p-value< 1e-03). As we have fitted mixed linear model between markers and phenotype and we know that in MLM R2 value does not play a major role. Also markers are correlated to each other causing multicollinearity leads to inflation in R2 value. This is a general trend in case of plant science where one can find such type of results. There are other studies reported earlier where marker R2 is very low but with low p-value hence contributing to phenotypes (Zegeye et al. 2014; Gyawali et al. 2017; Liu et al. 2017; Muleta et al. 2017; Ye et al. 2019)

6. PLOS authors have the option to publish the peer review history of their article (what does this mean?). If published, this will include your full peer review and any attached files.

Do you want your identity to be public for this peer review? For information about this choice, including consent withdrawal, please see our Privacy Policy.

Reviewer #1: Yes: Zhiyong Liu

Reviewer #2: No

---

## [Decision Letter · Decision Letter 1]

13 Nov 2019

PONE-D-19-19629R1

Genetic Characterization for Lesion mimic and other Traits in Relation to Spot blotch Resistance in Spring Wheat

PLOS ONE

Dear Dr. Mishra,

Thank you for submitting your manuscript to PLOS ONE. After careful consideration, we feel that it has merit but does not fully meet PLOS ONE’s publication criteria as it currently stands. Therefore, we invite you to submit a revised version of the manuscript that addresses the points raised during the review process.

ACADEMIC EDITOR: **Recommendations for decision are rather split. However, as one of reviewers (#2) pointed out, I think that this manuscript needs additional works. Therefore, I encourage the authors to perform the revision as per the reviewer’s suggestions and if these were meticulously performed, then I am sure that the MS could be reconsidered on a later date. **

We would appreciate receiving your revised manuscript by Dec 28 2019 11:59PM. To enhance the reproducibility of your results, we recommend that if applicable you deposit your laboratory protocols in protocols.io, where a protocol can be assigned its own identifier (DOI) such that it can be cited independently in the future. For instructions see: http://journals.plos.org/plosone/s/submission-guidelines#loc-laboratory-protocols

We look forward to receiving your revised manuscript.

Kind regards,

Manoj Prasad, PhD

Academic Editor

PLOS ONE

Reviewers' comments:

Reviewer's Responses to Questions

**Comments to the Author**

1. If the authors have adequately addressed your comments raised in a previous round of review and you feel that this manuscript is now acceptable for publication, you may indicate that here to bypass the “Comments to the Author” section, enter your conflict of interest statement in the “Confidential to Editor” section, and submit your "Accept" recommendation.

Reviewer #1: All comments have been addressed

Reviewer #2: (No Response)

2. Is the manuscript technically sound, and do the data support the conclusions?

Reviewer #1: Yes

Reviewer #2: Yes

3. Has the statistical analysis been performed appropriately and rigorously? 

Reviewer #1: Yes

Reviewer #2: Yes

4. Have the authors made all data underlying the findings in their manuscript fully available?

Reviewer #1: Yes

Reviewer #2: No

5. Is the manuscript presented in an intelligible fashion and written in standard English?

Reviewer #1: Yes

Reviewer #2: No

6. Review Comments to the Author

Reviewer #1: The authors revised the manuscript based on the reviewers' comments and suggestions. The quality of the manuscript was improved and I have no further comments.

Reviewer #2: I have acted as Reviewer #2 in the previous round of assessment of the manuscript. I have now focused on modifications made by the authors on the basis of the comments I have made. There are still a several issues to address, I indicate them below.

1) Rationale of the choice of varieties

Reviewer comment: this comment has not been addressed in the revised manuscript. Again, the comment was about the rationale of the choice of the varieties. This should be described.

2) Comments on further comments:

L60: Ltn was spelled out, but with about 5 edit mistakes: this should be: ". Another trait, leaf tip necrosis (Ltn), "

L63: the comment was addressed, but with mistakes: please edit and use the correct latin name for pathogen causing leaf rust and stripe rust

L65: I was referring to giving the full name instead of B. in B. sorokiniana; please provide the full name

L65-7: I did not see any change in the revised manuscript, in spite of the authors indicating otherwise in there answer to my comment.

L70: the phrase has been changed, but now has additional edit mistakes (spaces missing)

L70-2: although the sentence was rephrased, I still do not see the logics. Please rephrase. Do you mean that QTLs for resistance against head blight and stem rust were identified, and associated to Lm, while QTLs have not been identified in the case of resistance against spot blotch associated to Lm? What is also is not indicated, and what should be clearly stated, is if Lm has been associated to resistance against spot blotch on a phenotypic basis.

Title: "Scoring .." remove "disease" because this is addressed in the next sub-section

Assessments: additional information is needed:

- For Ltn, on what leaves were the assessments made?

- For glaucousness, a description of the 5 classes needs to be included.

- describe how LP was measured, from what observations; what was the frequency of observation? a reference is not sufficient

L267: I insist, this statement is not correct. It should be changed as for example: "The results of this study showed that higher LP is associated to lower levels of spot blotch and Lm." Looking at Table 2, the R2 values between GI and AUDPC and Lm are very small and should not in my view be emphasised in the Discussion.

3) Editing the text

(1) Many occurrences where a space is needed; this needs to be corrected throughout the text. (2) There are still many editorial mistakes which need to be addressed throughout the text.

7. PLOS authors have the option to publish the peer review history of their article (what does this mean?). If published, this will include your full peer review and any attached files.

Reviewer #1: No

Reviewer #2: No

---

## [Author Response · Author response to Decision Letter 1]

31 Dec 2019

ONE-D-19-19629

Genetic Characterization for Lesion mimic and other Traits in Relation to Spot blotch Resistance in Spring Wheat

PLOS ONE

Dear Dr. Manoj,

I am grateful you for giving me a chance to clarify the queries for the improving this paper as per standard of reputed journal ‘PLOS ONE’. I am also thankful to the reviewers who devoted much time and theirs inputs definitely improved the quality of paper. I hope you and reviewers will be satisfied with my inputs for each of your queries given here.

Reviewers' comments:

Reviewer's Responses to Questions

Comments to the Author

1. If the authors have adequately addressed your comments raised in a previous round of review and you feel that this manuscript is now acceptable for publication, you may indicate that here to bypass the “Comments to the Author”section, enter your conflict of interest statement in the “Confidential to Editor” section, and submit your "Accept" recommendation.

Reviewer #1: All comments have been addressed

Reviewer #2: (No Response)

2. Is the manuscript technically sounds, and do the data support the conclusions?

Reviewer #1: Yes

Reviewer #2: Yes

3. Has the statistical analysis been performed appropriately and rigorously?

Reviewer #1: Yes

Reviewer #2: Yes

4. Have the authors made all data underlying the findings in their manuscript fully available?

The PLOS Data policy requires authors to make all data underlying the findings described in their manuscript fully available without restriction, with rare exception (please refer to the Data Availability Statement in the manuscript PDF file). The data should be provided as part of the manuscript or its supporting information, or deposited to a public repository. For example, in addition to summary statistics, the data points behind means, medians and variance measures

should be available. If there are restrictions on publicly sharing data—e.g. participant privacy or use of data from a third party—those must be specified.

Reviewer #1: Yes

Reviewer #2: No

5. Is the manuscript presented in an intelligible fashion and written in standard English?

Reviewer #1: Yes

Reviewer #2: No

6. Review Comments to the Author

Reviewer #1: The authors revised the manuscript based on the reviewers' comments and suggestions. The quality of the manuscript was improved and I have no further comments.

Reviewer #2: I have acted as Reviewer #2 in the previous round of assessment of the manuscript. I have now focused on modifications made by the authors on the basis of the comments I have made. There are still a several issues to address, I indicate them below.

1) Rationale of the choice of varieties

Reviewer comment: this comment has not been addressed in the revised manuscript. Again, the comment was about the rationale of the choice of the varieties. This should be described.

Reply: Added in manuscript in material and method section. We chose the (Wheat Association Mapping) WAM population (CIMMYT Collection) for our study because it is a set of 289 diverse germplasm containing considerable genotypic and phenotypic variations. Since these are germplasm stocks so characters chosen for study are stable on both genotypic and phenotypic level, so in general there is no question of segregation or elimination of traits under examine.

2) Comments on further comments:

L60: Ltn was spelled out, but with about 5 edit mistakes: this should be: ". Another trait, leaf tip necrosis (Ltn), "

Reply: Agreed. Suggestion incorporated.

L63: the comment was addressed, but with mistakes: please edit and use the correct latin name for pathogen causing leaf rust and stripe rust.

Reply: Agreed. Mistakes were edited and corrected.

L65: I was referring to giving the full name instead of B. in B. sorokiniana; please provide the full name.

Reply: Agreed, full name was added as Bipolaris sorokiniana instead of B. sorokiniana.

L65-7: I did not see any change in the revised manuscript, in spite of the authors indicating otherwise in there answer to my comment.

Reply: Agreed. Explained in detail.

L70: The phrase has been changed, but now has additional edit mistakes (spaces missing).

Reply: Agreed, correction incorporated.

L70-2: although the sentence was rephrased, I still do not see the logics. Please rephrase. Do you mean that QTLs for resistance against head blight and stem rust were identified, and associated to Lm, while QTLs have not been identified in the case of resistance against spot blotch associated to Lm? What is also is not indicated, and what should be clearly stated, is if Lm has been associated to resistance against spot blotch on a phenotypic basis.

Reply: Here the sentence stated that “QTLs for resistance against head blight and stem rust were identified, and were associated to Lm”, The QTLs studies were carried out for Lm against these diseases due to protective nature of Lm (has been associated to resistance against these disease on a phenotypic basis) means they were negatively associated with Lm. While The present study we are describing in text, shown the positive association of Lm with Spot blotch, also there is no such QTLs reported in the case of spot blotch associated to Lm. This study was carried out to study the effect of Lm on spot blotch.

Title: "Scoring .." remove "disease" because this is addressed in the next sub-section

Assessments: additional information is needed:

Reply: Agreed, correction incorporated according to suggestions.

- For Ltn, on what leaves were the assessments made?

Reply: It was scored on flag leaves. (Now mentioned in concern Para).

- For glaucousness, a description of the 5 classes needs to be included.

Reply: Agreed, scale described in related Para.

- describe how LP was measured, from what observations; what was the frequency of observation? a reference is not sufficient.

Reply: Description added in concern Para.

L267: I insist, this statement is not correct. It should be changed as for example: "The results of this study showed that higher LP is associated to lower levels of spot blotch and Lm." 

Reply: Agreed, sentence modified according to suggestion.

Looking at Table 2, the R2 values between GI and AUDPC

and Lm are very small and should not in my view be emphasised in the Discussion.

Reply: R values are small but significant hence discussed. 

3) Editing the text

(1) Many occurrences where a space is needed; this needs to be corrected throughout the text. 

Reply: Correction made throughout the text and tries best to rectify it.

(2) There are still many editorial mistakes which need to be addressed throughout the text.

Reply: Agreed and mistakes were corrected in main text. 

7. PLOS authors have the option to publish the peer review history of their article (what does this mean?). If published, this will include your full peer review and any attached files.

Do you want your identity to be public for this peer review? For information about this choice, including consent

withdrawal, please see our Privacy Policy.

Reviewer #1: No

Reviewer #2: No

While revising your submission, please upload your figure files to the Preflight Analysis and Conversion Engine (PACE)

digital diagnostic tool, https://pacev2.apexcovantage.com/. PACE helps ensure that figures meet PLOS requirements. To

use PACE, you must first register as a user. Registration is free. Then, login and navigate to the UPLOAD tab, where you

will find detailed instructions on how to use the tool. If you encounter any issues or have any questions when using PACE,

please email us at figures@plos.org. Please note that Supporting Information files do not need this step.

In compliance with data protection regulations, you may request that we remove your personal registration details at any time. (Remove my

information/details). Please contact the publication office if you have any questions.

---

## [Decision Letter · Decision Letter 2]

22 Jan 2020

PONE-D-19-19629R2

Genetic Characterization for Lesion mimic and other Traits in Relation to Spot blotch Resistance in Spring Wheat

PLOS ONE

Dear Dr. Mishra,

Thank you for submitting your manuscript to PLOS ONE. After careful consideration, we feel that it has merit but does not fully meet PLOS ONE’s publication criteria as it currently stands. Therefore, we invite you to submit a revised version of the manuscript that addresses the points raised during the review process.

ACADEMIC EDITOR: **The revised version is now much improved but still there are several issues as indicated by the reviewer number 2. Therefore, authors are advised to revise their manuscript following reviewer comments.**

We would appreciate receiving your revised manuscript by Mar 07 2020 11:59PM. To enhance the reproducibility of your results, we recommend that if applicable you deposit your laboratory protocols in protocols.io, where a protocol can be assigned its own identifier (DOI) such that it can be cited independently in the future. For instructions see: http://journals.plos.org/plosone/s/submission-guidelines#loc-laboratory-protocols

We look forward to receiving your revised manuscript.

Kind regards,

Manoj Prasad, PhD

Academic Editor

PLOS ONE

Reviewers' comments:

Reviewer's Responses to Questions

**Comments to the Author**

1. If the authors have adequately addressed your comments raised in a previous round of review and you feel that this manuscript is now acceptable for publication, you may indicate that here to bypass the “Comments to the Author” section, enter your conflict of interest statement in the “Confidential to Editor” section, and submit your "Accept" recommendation.

Reviewer #1: All comments have been addressed

Reviewer #2: (No Response)

2. Is the manuscript technically sound, and do the data support the conclusions?

Reviewer #1: Yes

Reviewer #2: Yes

3. Has the statistical analysis been performed appropriately and rigorously? 

Reviewer #1: Yes

Reviewer #2: Yes

4. Have the authors made all data underlying the findings in their manuscript fully available?

Reviewer #1: Yes

Reviewer #2: Yes

5. Is the manuscript presented in an intelligible fashion and written in standard English?

Reviewer #1: Yes

Reviewer #2: No

6. Review Comments to the Author

Reviewer #1: The authors addressed my concerns and I have no further comment! The manuscript was revised again based on another reviewers comment and the entire manuscript was improved.

Reviewer #2: I have acted as Reviewer #2 in the previous round of assessment of the manuscript. I have now focused on modifications made by the authors on the basis of the comments I have made. There are still a few issues to address, I indicate them below. The main issue I see, for which time will be required, is with respect to editing the English.

1) Rationale of the choice of varieties

This comment is now accounted for but further editing is needed. I suggest to replace the sentence "The population contains elite genetically diverse germplasm and so are stable for the traits under examine on both genotypic and phenotypic level and in general ruled out the chance of any segregation or elimination of charecters at maximum extent."

By the following sentence:

"The population contains elite germplasm displaying considerable genetic and phenotypic variation. Furthermore, the population includes genotypes which are stable for the traits under examination."

2) Further comments:

The whole document should be revised thoroughly for editing, and to remove mistakes (e.g., space missing between two consecutive words). I am indicating additional specific comments are below:

L61: “Another trait, Leaf tip necrosis (Ltn),”: remove the symbols " at the beginning and at the end of this segment

L67: "resistance against spot blotch caused by Bipolaris sorokiniana ": add "the hemibiotroph pathogen" after "caused by"

L68: add "increased" before "latent"

L79-89: there are several problems here. One of them is that Lm is not associated to resistance vs fusarium head blight: the sentence needs to be changed and the reference 27 removed; FHB is caused by Fusarium spp. which are not biotrophic. Also, there are many sentences which repeat what was already written previously, and so which need to be removed. I propose to change this paragraph as follows:

"Associations between Lm, Ltn, GI, and LP, and spot blotch have not been studied."

This sentence is to be followed by the beginning of the next paragraph.

L94: add " Ltn, GI, and LP" after "Lm"

L94: replace "its" by "their"

L123: remove "and disease"

L130-131: delete "was recorded in the field"

L370: "LP also showed negative effect on Lm. Longer LP was not found favorable for the expression of Lm and contributed to enhance resistance against spot blotch": this needs to be changed: LP cannot be the cause for Lm.

Please replace by "LP was negatively associated with Lm. Longer LP contributed to enhance resistance against spot blotch"

L380-2: this needs rephrasing for English and for the meaning.

7. PLOS authors have the option to publish the peer review history of their article (what does this mean?). If published, this will include your full peer review and any attached files.

Reviewer #1: Yes: Zhiyong Liu

Reviewer #2: No

---

## [Author Response · Author response to Decision Letter 2]

28 Feb 2020

ONE-D-19-19629

Genetic Characterization for Lesion mimic and other Traits in Relation to Spot blotch Resistance in Spring Wheat

PLOS ONE

Dear Dr. Manoj,

I am grateful you for giving me a chance to clarify the queries for the improving this paper as per standard of reputed journal ‘PLOS ONE’. I am also thankful to the reviewers who devoted much time and theirs inputs definitely improved the quality of paper. I hope you and reviewers will be satisfied with my inputs for each of your queries given here.

Reviewers' comments:

Reviewer's Responses to Questions

Comments to the Author

1. If the authors have adequately addressed your comments raised in a previous round of review and you feel that this

manuscript is now acceptable for publication, you may indicate that here to bypass the “Comments to the Author”

section, enter your conflict of interest statement in the “Confidential to Editor” section, and submit your "Accept"

recommendation.

Reviewer #1: All comments have been addressed

Reviewer #2: (No Response)

2. Is the manuscript technically sound, and do the data support the conclusions?

Reviewer #1: Yes

Reviewer #2: Yes

3. Has the statistical analysis been performed appropriately and rigorously?

Reviewer #1: Yes

Reviewer #2: Yes

4. Have the authors made all data underlying the findings in their manuscript fully available?

Reviewer #1: Yes

Reviewer #2: Yes

5. Is the manuscript presented in an intelligible fashion and written in standard English?

Reviewer #1: Yes

Reviewer #2: No

6. Review Comments to the Author

Reviewer #1: The authors addressed my concerns and I have no further comment! The manuscript was revised again based on another reviewers comment and the entire manuscript was improved.

Reviewer #2: I have acted as Reviewer #2 in the previous round of assessment of the manuscript. I have now focused on modifications made by the authors on the basis of the comments I have made. There are still a few issues to address, I indicate them below. The main issue I see, for which time will be required, is with respect to editing the English.

1) Rationale of the choice of varieties

This comment is now accounted for but further editing is needed. I suggest to replace the sentence "The population contains elite genetically diverse germplasm and so are stable for the traits under examine on both genotypic and phenotypic level and in general ruled out the chance of any segregation or elimination of charecters at maximum extent."

By the following sentence: "The population contains elite germplasm displaying considerable genetic and phenotypic variation. Furthermore, the population includes genotypes which are stable for the traits under examination."

Reply: Agreed, Thanks for your kind suggestions, correction incorporated according to suggestions.

2) Further comments:

The whole document should be revised thoroughly for editing, and to remove mistakes (e.g., space missing between two consecutive words).

Reply: Agreed, it was due to different version of Microsoft word office.

 I am indicating additional specific comments are below:

L61: “Another trait, Leaf tip necrosis (Ltn),”: remove the symbols " at the beginning and at the end of this segment.

Reply: Agreed, correction incorporated according to suggestions.

L67: "resistance against spot blotch caused by Bipolaris sorokiniana ": add "the hemibiotroph pathogen" after "caused by"

Reply: Agreed, correction incorporated according to suggestions.

L68: add "increased" before "latent"

Reply: Agreed, correction incorporated according to suggestions.

L79-89: there are several problems here. One of them is that Lm is not associated to resistance vs fusarium head blight: the sentence needs to be changed and the reference 27 removed; FHB is caused by Fusarium spp. which are not biotrophic. Also, there are many sentences which repeat what was already written previously, and so which need to be removed. I propose to change this paragraph as follows: "Associations between Lm, Ltn, GI, and LP, and spot blotch have not been studied."

This sentence is to be followed by the beginning of the next paragraph.

Reply: Agreed, reference no 27 removed and sentence reframed according to suggestions.

L94: add " Ltn, GI, and LP" after "Lm"

Reply: Agreed, correction incorporated according to suggestions.

L94: replace "its" by "their"

Reply: Agreed, word replaced according to suggestions.

L123: remove "and disease"

Reply: Agreed, word removed in main text.

L130-131: delete "was recorded in the field"

Reply: Agreed, was deleted in main text.

L370: "LP also showed negative effect on Lm. Longer LP was not found favorable for the expression of Lm and contributed to enhance resistance against spot blotch": this needs to be changed: LP cannot be the cause for Lm. Please replace by "LP was negatively associated with Lm. Longer LP contributed to enhance resistance against spot blotch"

Reply: Agreed, correction incorporated according to suggestions.

L380-2: this needs rephrasing for English and for the meaning.

Reply: Agreed, correction incorporated and paragraph rephrased according to suggestions.

7. PLOS authors have the option to publish the peer review history of their article (what does this mean?). If published,

this will include your full peer review and any attached files.

Do you want your identity to be public for this peer review? For information about this choice, including consent

withdrawal, please see our Privacy Policy.

Reviewer #1: Yes: Zhiyong Liu

Reviewer #2: No

While revising your submission, please upload your figure files to the Preflight Analysis and Conversion Engine (PACE)

digital diagnostic tool, https://pacev2.apexcovantage.com/. PACE helps ensure that figures meet PLOS requirements.

To use PACE, you must first register as a user. Registration is free. Then, login and navigate to the UPLOAD tab, where

you will find detailed instructions on how to use the tool. If you encounter any issues or have any questions when

using PACE, please email us at figures@plos.org. Please note that Supporting Information files do not need this step.

In compliance with data pr otection regulations, y ou may request

---

## [Decision Letter · Decision Letter 3]

9 Apr 2020

PONE-D-19-19629R3

Genetic Characterization for Lesion mimic and other Traits in Relation to Spot blotch Resistance in Spring Wheat

PLOS ONE

Dear Dr. Mishra,

Thank you for submitting your manuscript to PLOS ONE. After careful consideration, we feel that it has merit but does not fully meet PLOS ONE’s publication criteria as it currently stands. Therefore, we invite you to submit a revised version of the manuscript that addresses the points raised during the review process.

The revised manuscript is now much improved but still there are several problems as pointed out by both the reviewers. Therefore, authors are advised to revise their manuscript following reviewers comments. I would further request the authors to answer the reviewers (particularly reviewer number 4) questions seriously.

We would appreciate receiving your revised manuscript by May 24 2020 11:59PM. To enhance the reproducibility of your results, we recommend that if applicable you deposit your laboratory protocols in protocols.io, where a protocol can be assigned its own identifier (DOI) such that it can be cited independently in the future. For instructions see: http://journals.plos.org/plosone/s/submission-guidelines#loc-laboratory-protocols

We look forward to receiving your revised manuscript.

Kind regards,

Manoj Prasad, PhD

Academic Editor

PLOS ONE

Reviewers' comments:

Reviewer's Responses to Questions

**Comments to the Author**

1. If the authors have adequately addressed your comments raised in a previous round of review and you feel that this manuscript is now acceptable for publication, you may indicate that here to bypass the “Comments to the Author” section, enter your conflict of interest statement in the “Confidential to Editor” section, and submit your "Accept" recommendation.

Reviewer #3: All comments have been addressed

Reviewer #4: (No Response)

2. Is the manuscript technically sound, and do the data support the conclusions?

Reviewer #3: Yes

Reviewer #4: Yes

3. Has the statistical analysis been performed appropriately and rigorously? 

Reviewer #3: Yes

Reviewer #4: Yes

4. Have the authors made all data underlying the findings in their manuscript fully available?

Reviewer #3: Yes

Reviewer #4: Yes

5. Is the manuscript presented in an intelligible fashion and written in standard English?

Reviewer #3: No

Reviewer #4: No

6. Review Comments to the Author

Reviewer #3: The study defines the genetic characterization of Lm, Ltn, GI, LP, and their association with spot blotch resistance in spring wheat. I could see that the manuscript was previously reviewed by two reviewers who had already commented twice and based on those comments, the manuscript was improved. I have gone through all the versions of the manuscript, reviewers’ comments and the response of authors. The work is scientifically robust and technically sound, as major flaws were addressed during the revisions. However, the authors were repeatedly requested to improve the language, which they were apparently not able to. There are several minor and major issues pertaining to sentence formation, choice of vocabulary, voice agreement and use/misuse of punctuation marks. Plos One does not copyedit accepted manuscripts, and therefore, the language in submitted articles must be clear, correct, and unambiguous. I strongly recommend the authors to take assistance from a professional editing service to resolve the language issues. The certificate stating that the language was professionally edited should also be submitted to the journal for validation.

Reviewer #4: The manuscript is not well written. It needs language and scientific editing. I have highlighted some sentences in abstract that needs attention. Similar problems have been noticed i other portions of the manuscript. I think manuscript has good potential for publication in PLOS ONE but authors shall edit whole manuscript or get the manuscript edited from someone familiar with GWAS.

My feedback for Abstract:

1. The sentence < Lesion mimic (Lm) mutants are hypersensitive responses (HR) phenotype in the absence of the pathogen.> should be rewritten to make meaning clear

2. In sentence < In wheat, such mutants were reported to be resistant against rustsdue to their biotrophic nature.> the authors shall mention clearly which rust because there are three common rust disease in wheat .

3. In sentence < In this study, 289 diverse wheat germplasm were phenotyped for 3 consecutive years (2012 to 2015).> the author shall write 289 genotypes or 289 wheat germplasm lines

4. The sentence < Genotyping was done using Illumina iSelect beadchip assay for wheat having13589 highly polymorphic SNPs used for association mapping.> does not read well

5. The authors wrote < In genome wise association study (GWAS),> but it is < In genome wide association study (GWAS),>

6. < associated to various traits> may be changed with < associated with various traits>

similar problems have been noticed by me in all sections of the manuscript.

General Comments:

1. The manuscript is not well written. It needs both language editing and scientific editing.

2. The results of GWAS are not intelligently presented. There is no mention about contribution of associated markers although same is presented in Table.

3. In structural analysis there is no mention about admixed genotypes although same could be seen in structural plot

4. The quality of figures is poor

5. Manhattan plot are poorly presented. The authors shall highlight the significant SNPs in Manhattan plots

6. The results of ANOVA are not properly discussed

7. PLOS authors have the option to publish the peer review history of their article (what does this mean?). If published, this will include your full peer review and any attached files.

Reviewer #3: No

Reviewer #4: No

---

## [Author Response · Author response to Decision Letter 3]

3 Jun 2020

Comments to the Author

1. If the authors have adequately addressed your comments raised in a previous round of review and you feel that this manuscript is now acceptable for publication, you may indicate that here to bypass the “Comments to the Author” section, enter your conflict of interest statement in the “Confidential to Editor” section, and submit your "Accept" recommendation.

Reviewer #3: All comments have been addressed

Reviewer #4: (No Response)

2. Is the manuscript technically sound, and do the data support the conclusions?

Reviewer #3: Yes

Reviewer #4: Yes

3. Has the statistical analysis been performed appropriately and rigorously?

Reviewer #3: Yes

Reviewer #4: Yes

4. Have the authors made all data underlying the findings in their manuscript fully available?

Reviewer #3: Yes

Reviewer #4: Yes

5. Is the manuscript presented in an intelligible fashion and written in standard English?

Reviewer #3: No

Reviewer #4: No

6. Review Comments to the Author

Reviewer #3: The study defines the genetic characterization of Lm, Ltn, GI, LP, and their association with spot blotch resistance in spring wheat. I could see that the manuscript was previously reviewed by two reviewers who had already commented twice and based on those comments, the manuscript was improved. I have gone through all the versions of the manuscript, reviewers’ comments and the response of authors. The work is scientifically robust and technically sound, as major flaws were addressed during the revisions. However, the authors were repeatedly requested to improve the language, which they were apparently not able to. There are several minor and major issues pertaining to sentence formation, choice of vocabulary, voice agreement and use/misuse of punctuation marks. Plos One does not copyedit accepted manuscripts, and therefore, the language in submitted articles must be clear, correct, and unambiguous. I strongly recommend the authors to take assistance from a professional editing service to resolve the language issues.

The certificate stating that the language was professionally edited should also be submitted to the journal for validation.

Reviewer #4: The manuscript is not well written. It needs language and scientific editing. I have highlighted some sentences in abstract that needs attention. Similar problems have been noticed i other portions of the manuscript. I think manuscript has good potential for publication in PLOS ONE but authors shall edit whole manuscript or get the manuscript edited from someone familiar with GWAS.

My feedback for Abstract:

1. The sentence < Lesion mimic (Lm) mutants are hypersensitive responses (HR) phenotype in the absence of the pathogen.> should be rewritten to make meaning clear

Reply: Agreed, Changed the sentence accordingly.

2. In sentence < In wheat, such mutants were reported to be resistant against rusts due to their biotrophic nature.> the authors shall mention clearly which rust because there are three common rust disease in wheat.

Reply: Agreed, added in abstract.

3. In sentence < In this study, 289 diverse wheat germplasm were phenotyped for 3 consecutive years (2012 to 2015).> the author shall write 289 genotypes or 289 wheat germplasm lines.

Reply: Agreed, sentence was modified in main document, thanks for your kind suggestion. 

3. The sentence < Genotyping was done using Illumina iSelect beadchip assay for wheat having13589 highly polymorphic SNPs used for association mapping.> does not read well.

Reply: Agreed, correction incorporated

4. The authors wrote < In genome wise association study (GWAS),> but it is < In genome wide association study (GWAS),>

Reply: Agreed, correction incorporated. 

6. < associated to various traits> may be changed with < associated with various traits>

similar problems have been noticed by me in all sections of the manuscript.

Reply: Agreed, correction incorporated. 

General Comments:

1. The manuscript is not well written. It needs both language editing and scientific editing.

Reply: A thorough English and science editing was done using co-authors and a professional Science Editor

2. The results of GWAS are not intelligently presented. There is no mention about contribution of associated markers although same is presented in Table.

Reply: Agreed, desired is incorporated.

3. In structural analysis there is no mention about admixed genotypes although same could be seen in structural plot.

Reply: Said information is incorporated in the manuscript.

4. The quality of figures is poor

Reply: Agreed, quality of figure is improved upto 300 dpi.

5. Manhattan plot are poorly presented. The authors shall highlight the significant SNPs in Manhattan plots.

Reply: Quality of Manhattan plot is improved, a threshold line at (-log10(P-value)=3) is drawn to highlight significant markers.

6. The results of ANOVA are not properly discussed

Reply: Agreed, results of ANOVA has been discussed in detail.

7. PLOS authors have the option to publish the peer review history of their article (what does this mean?). If published,

this will include your full peer review and any attached files.

Do you want your identity to be public for this peer review? For information about this choice, including consent

withdrawal, please see our Privacy Policy.

Reviewer #3: No

Reviewer #4: No

---

## [Decision Letter · Decision Letter 4]

9 Jul 2020

PONE-D-19-19629R4

Genetic characterization for lesion mimic and other traits in relation to spot blotch resistance in spring wheat

PLOS ONE

Dear Dr. Mishra,

Thank you for submitting your manuscript to PLOS ONE. After careful consideration, we feel that it has merit but does not fully meet PLOS ONE’s publication criteria as it currently stands. Therefore, we invite you to submit a revised version of the manuscript that addresses the points raised during the review process.

ACADEMIC EDITOR:

The reviewers' comments are back on your ms which shows that the work shall be revised and resubmitted. Therefore, authors are advised to modify and submit their work accordingly.

We look forward to receiving your revised manuscript.

Kind regards,

Manoj Prasad, PhD

Academic Editor

PLOS ONE

Reviewers' comments:

Reviewer's Responses to Questions

**Comments to the Author**

1. If the authors have adequately addressed your comments raised in a previous round of review and you feel that this manuscript is now acceptable for publication, you may indicate that here to bypass the “Comments to the Author” section, enter your conflict of interest statement in the “Confidential to Editor” section, and submit your "Accept" recommendation.

Reviewer #3: All comments have been addressed

Reviewer #4: All comments have been addressed

2. Is the manuscript technically sound, and do the data support the conclusions?

Reviewer #3: Yes

Reviewer #4: Yes

3. Has the statistical analysis been performed appropriately and rigorously? 

Reviewer #3: N/A

Reviewer #4: Yes

4. Have the authors made all data underlying the findings in their manuscript fully available?

Reviewer #3: Yes

Reviewer #4: Yes

5. Is the manuscript presented in an intelligible fashion and written in standard English?

Reviewer #3: Yes

Reviewer #4: Yes

6. Review Comments to the Author

Reviewer #3: Authors have addressed all the comments raised by the reviewers, and there is no further revisions required.

Reviewer #4: The manuscript has been improved by keeping in view the comments made by reviewers but I think there are still language issues in the manuscript. For instance In material and methods (Plant material section) LN 109-114 the authors are using past tense for <wami obtained="" population="" were=""> as well future tense <to be="" for="" study="" this="" used="">. The authors need to maintain consistency throughout manuscript. In the same section the authors are still using <289 diverse wheat germplasm> despite suggestions were provided in previous revision. In addition I noticed that in table # 3 for LM, Gi etc the different associated markers are on same chromosomes with same position. Can authors have a look on the raw data again because having same position may signal same gene/QTL and not different genes/QTLs on that particular chromosome.</to></wami>

7. PLOS authors have the option to publish the peer review history of their article (what does this mean?). If published, this will include your full peer review and any attached files.

Reviewer #3: No

Reviewer #4: No

---

## [Author Response · Author response to Decision Letter 4]

22 Aug 2020

PONE-D-19-19629R3

Genetic Characterization for Lesion mimic and other Traits in Relation to Spot blotch Resistance in Spring Wheat

Responses to Reviewers comments

Comments to the Author

1. If the authors have adequately addressed your comments raised in a previous round of review and you feel that this manuscript is now acceptable for publication, you may indicate that here to bypass the “Comments to the Author” section, enter your conflict of interest statement in the “Confidential to Editor” section, and submit your "Accept"

recommendation.

Reviewer #3: All comments have been addressed

Reviewer #4: All comments have been addressed

2. Is the manuscript technically sound, and do the data support the conclusions?

Reviewer #3: Yes

Reviewer #4: Yes

3. Has the statistical analysis been performed appropriately and rigorously?

Reviewer #3: N/A

Reviewer #4: Yes

4. Have the authors made all data underlying the findings in their manuscript fully available?

The PLOS Data policy requires authors to make all data underlying the findings described in their manuscript fully available without restriction, with rare exception (please refer to the Data Availability Statement in the manuscript PDF file). The data should be provided as part of the manuscript or its supporting information, or deposited to a public repository. For example, in addition to summary statistics, the data points behind means, medians and variance measures should be available. If there are restrictions on publicly sharing data—e.g. participant privacy or use of data from a third

party—those must be specified.

Reviewer #3: Yes

Reviewer #4: Yes

5. Is the manuscript presented in an intelligible fashion and written in standard English?

Reviewer #3: Yes

Reviewer #4: Yes

6. Review Comments to the Author

Reviewer #3: Authors have addressed all the comments raised by the reviewers, and there is no further revisions required.

Reply: Authors are the thankful for the positive feedback.

Reviewer #4: The manuscript has been improved by keeping in view the comments made by reviewers but I think there are still language issues in the manuscript. 

For instance In material and methods (Plant material section) LN 109-114 the authors are using past tense for as well future tense . The authors need to maintain consistency throughout manuscript. 

Reply: Agreed, the sentence improved accordingly and manuscript too. A thorough English and science editing was done using co-authors and a professional Science Edito. The certificate from professional writer is attached.

In the same section the authors are still using <289 diverse wheat germplasm> despite suggestions were provided in previous revision. 

Reply: Agreed, the sentence improved accordingly and manuscript too.

In addition I noticed that in table # 3 for LM, Gi etc the different associated markers are on same chromosomes with same position. Can authors have a look on the raw data again because having same position may signal same gene/QTL and not different genes/QTLs on that particular chromosome.

Reply: Thanks for this good observation. However we have gone through raw data also and found that these are different SNPs (mapped at same genetic distance i.e. cM) and we

have also confirmed these results from earlier studies where multiple SNPs are located at same genetic distance (this may be due to reason that cM distance represents large chunks of genomic regions, sometimes in hundreds of KB).

 However we have highlighted those SNPs which represent the same candidate genes in the annotation section of manuscript.

7. PLOS authors have the option to publish the peer review history of their article (what does this mean?). If published, this will include your full peer review and any attached files.

Do you want your identity to be public for this peer review? 

Reply: No.

---

## [Decision Letter · Decision Letter 5]

18 Sep 2020

Genetic characterization for lesion mimic and other traits in relation to spot blotch resistance in spring wheat

PONE-D-19-19629R5

Dear Dr. Mishra,

We’re pleased to inform you that your manuscript has been judged scientifically suitable for publication and will be formally accepted for publication once it meets all outstanding technical requirements.

Kind regards,

Manoj Prasad, PhD

Academic Editor

PLOS ONE

Additional Editor Comments (optional):

Reviewers' comments:

Reviewer's Responses to Questions

**Comments to the Author**

1. If the authors have adequately addressed your comments raised in a previous round of review and you feel that this manuscript is now acceptable for publication, you may indicate that here to bypass the “Comments to the Author” section, enter your conflict of interest statement in the “Confidential to Editor” section, and submit your "Accept" recommendation.

Reviewer #3: All comments have been addressed

Reviewer #4: All comments have been addressed

2. Is the manuscript technically sound, and do the data support the conclusions?

Reviewer #3: Yes

Reviewer #4: Yes

3. Has the statistical analysis been performed appropriately and rigorously? 

Reviewer #3: N/A

Reviewer #4: Yes

4. Have the authors made all data underlying the findings in their manuscript fully available?

Reviewer #3: Yes

Reviewer #4: (No Response)

5. Is the manuscript presented in an intelligible fashion and written in standard English?

Reviewer #3: Yes

Reviewer #4: Yes

6. Review Comments to the Author

Reviewer #3: Authors have addressed all the comments raised by the reviewers, and there is no further revisions required.

Reviewer #4: (No Response)

7. PLOS authors have the option to publish the peer review history of their article (what does this mean?). If published, this will include your full peer review and any attached files.

Reviewer #3: No

Reviewer #4: No

---

## [Editor Report · Acceptance letter]

25 Sep 2020

PONE-D-19-19629R5 

Genetic characterization for lesion mimic and other traits in relation to spot blotch resistance in spring wheat 

Dear Dr. Mishra:

I'm pleased to inform you that your manuscript has been deemed suitable for publication in PLOS ONE. Congratulations! Your manuscript is now with our production department. 

Kind regards, 

on behalf of

Dr. Manoj Prasad 

Academic Editor

PLOS ONE